# Molecular Cytogenetics in Domestic Bovids: A Review

**DOI:** 10.3390/ani13050944

**Published:** 2023-03-06

**Authors:** Alessandra Iannuzzi, Leopoldo Iannuzzi, Pietro Parma

**Affiliations:** 1Institute for Animal Production System in Mediterranean Environment, National Research Council, 80055 Portici, Italy; 2Department of Agricultural and Environmental Sciences, University of Milan, 20133 Milan, Italy

**Keywords:** animal cytogenetics, cattle, river buffalo, sheep, goat, FISH mapping, PCR

## Abstract

**Simple Summary:**

Molecular cytogenetics, and particularly the use of fluorescence in situ hybridization (FISH), has allowed deeper investigation of the chromosomes of domestic animals in order to: (a) create physical maps of specific DNA sequences on chromosome regions; (b) use specific chromosome markers to confirm the identification of chromosomes or chromosome regions involved in chromosome abnormalities, especially when poor banding patterns are produced; (c) better anchor radiation hybrid and genetic maps to specific chromosome regions; (d) better compare related and unrelated species by comparative FISH mapping and/or Zoo-FISH techniques; (e) study meiotic segregation, especially by sperm-FISH, in some chromosome abnormalities; (f) better show conserved or lost DNA sequences in chromosome abnormalities; (g) use informatic and genomic reconstructions, in addition to CGH arrays in related species, to predict conserved or lost chromosome regions; and (h) study some chromosome abnormalities and genomic stability using PCR applications. This review summarizes the most important applications of molecular cytogenetics in domestic bovids, with an emphasis on FISH mapping applications.

**Abstract:**

The discovery of the Robertsonian translocation (rob) involving cattle chromosomes 1 and 29 and the demonstration of its deleterious effects on fertility focused the interest of many scientific groups on using chromosome banding techniques to reveal chromosome abnormalities and verify their effects on fertility in domestic animals. At the same time, comparative banding studies among various species of domestic or wild animals were found useful for delineating chromosome evolution among species. The advent of molecular cytogenetics, particularly the use of fluorescence in situ hybridization (FISH), has allowed a deeper investigation of the chromosomes of domestic animals through: (a) the physical mapping of specific DNA sequences on chromosome regions; (b) the use of specific chromosome markers for the identification of the chromosomes or chromosome regions involved in chromosome abnormalities, especially when poor banding patterns are produced; (c) better anchoring of radiation hybrid and genetic maps to specific chromosome regions; (d) better comparisons of related and unrelated species by comparative FISH mapping and/or Zoo-FISH techniques; (e) the study of meiotic segregation, especially by sperm-FISH, in some chromosome abnormalities; (f) better demonstration of conserved or lost DNA sequences in chromosome abnormalities; (g) the use of informatic and genomic reconstructions, in addition to CGH arrays, to predict conserved or lost chromosome regions in related species; and (h) the study of some chromosome abnormalities and genomic stability using PCR applications. This review summarizes the most important applications of molecular cytogenetics in domestic bovids, with an emphasis on FISH mapping applications.

## 1. Introduction

The application of cytogenetics to domestic animals emerged about 60 years ago with the study of normal stained chromosome preparations from some cases of domestic animals with reproductive defects [1,2,3]. However, the discovery of the Robertsonian translocation (rob) involving cattle chromosomes 1 and 29 [4,5] and the demonstration of its deleterious effects on fertility [6,7] was what piqued the interest of many scientific groups and focused their attention on studying the chromosomes of domestic animals. This approach was particularly useful for selecting bulls to be used for artificial insemination, as it could avoid the transmission of chromosome abnormalities (i.e., rob1;29) from bull carriers to their progeny. Evolutionary studies also benefitted from advancements beyond normal chromosome staining. Among the various studies, the most important was the study of the Bovidae family by Wurster and Benirske [8], who looked at the diploid number and shape of chromosomes. They concluded that while the diploid number varies from 38 to 60 among all bovid species, the number of chromosome arms (Fundamental Number = NF) varies only between 58 and 62, with three exceptions; therefore, they hypothesized a high degree of autosome arm conservation among all bovid species. This hypothesis was later confirmed with the application of chromosome banding techniques [9], which ushered in a new era of chromosome studies in various domestic animal species, allowing (a) the establishment of standard karyotypes of the most important domestic species as a point of reference for various applications; (b) better characterization and identification of the chromosomes involved in chromosome abnormalities of domestic animals [10], particularly domestic bovids [11], pigs [12], horses [13], and dogs [14]; (c) the study of the chromosome homologies between related and unrelated species [15,16,17]; and (d) the study of chromosome fragility in animals exposed in vivo or in vitro to particular mutagens [18,19]. The molecular cytogenetics, particularly the introduction of fluorescence in situ hybridization (FISH), offered a deeper investigation of the chromosomes of domestic animals through: (a) the physical mapping of specific DNA sequences on chromosome regions; (b) the use of specific chromosome markers for the identification of chromosomes or chromosome regions involved in chromosome abnormalities, especially when poor banding patterns are produced; (c) better anchoring of radiation hybrid (RH) and genetic maps to specific chromosome regions; (d) better comparisons of related and unrelated species by comparative FISH mapping and/or Zoo-FISH techniques; (e) the study of meiotic segregation, especially by sperm-FISH, in some chromosome abnormalities or aneuploidies in both oocytes and embryos; (f) better demonstration of conserved or lost DNA sequences in chromosome abnormalities by CGH (comparative genomic hybridization) or SNP (single-nucleotide polymorphism) arrays; (g) the use of informatic and genomic reconstructions, in addition to CGH arrays, for the prediction of conserved or lost chromosome regions in related species; and (h) the study of chromosome abnormalities and genomic stability using PCR (polymerase chain reaction).

This review summarizes the most important applications of molecular cytogenetics in domestic bovids, with particular emphasis on FISH mapping applications.

## 2. The Fluorescence In Situ Hybridization (FISH) Technique

The FISH mapping technique is based on two main principles: the target and the probe. The target can be a whole chromosome (or chromosome arms) or a specific chromosome region. The probe is prepared according to the size of the target and is typically: (a) cDNA (generally applied when the target gene is a multi-copy); (b) cosmids with DNA insert sizes of 20–40 kb; (c) bacterial artificial chromosomes (BACs) with DNA insert sizes of 100–300 kb; (d) yeast artificial chromosome (YAC) clones (these are actually not used because they have a low cloning efficiency and show a high level of chimerism); (e) chromosome painting probes (obtained by cell sorter or chromosome microdissection techniques) that can visualize parts of or entire chromosomes; and (f) CGH arrays to check for genomic gains or losses. The probes are labeled directly with fluorochromes or indirectly with molecules that bind to the probe via fluorochrome-conjugated antibodies. The probe is specific for the target, based on complementary DNA base pairing, which allows the fluorescence-labeled probes to hybridize and form specific fluorescent signals on specific chromosome regions.

The advent of the fluorescence in situ hybridization (FISH) technique, initially applied to human chromosomes [20,21], noticeably expanded cytogenetics research and investigations applied to domestic animals due to the possibility of revealing specific chromosome regions, entire chromosomes, or chromosome arms according to the choice of probe. One of the great advantages of the FISH technique is that it can be applied to interphase cell nuclei, meiotic preparations (sperm and oocytes), embryos, and elongated chromatin fibers, in addition to metaphase chromosomes, thereby allowing more complete cytogenetic investigations of animal cells. The following sections describe the main uses of FISH in domestic bovids.

### 2.1. FISH and Chromosome Abnormalities

The first study to apply FISH for the precise identification of the chromosomes involved in a chromosome abnormality was published by Gallagher et al. [22], who discovered an X-autosome translocation (X;23) using both Q-banding and a BoLA Class I cDNA probe. The probe shows hybridization signals to the normal chromosome 23 and to the translocated autosomal material present on the X chromosome, allowing a more precise localization of MHC (major histocompatibility complex) in cattle than was achieved earlier by genetic mapping. Several subsequent studies also applied FISH to obtain better confirmation of the chromosome(s) involved in abnormalities (especially when banding was poor) and identification of the break points, especially in reciprocal translocations. Table 1 shows the main studies that applied FISH mapping, either alone or in combination with other classical cytogenetic techniques (e.g., C-banding, G-banding, R-banding, and Ag-NORs), to study the chromosome abnormalities of domestic bovids in somatic cells at the metaphase (Figure 1) or interphase nuclei of germinal cells, such as sperm and oocytes, or embryos at different cell stages. 

A more complete classification of all chromosome abnormalities studied by classical cytogenetic techniques alone or (in some cases) with other molecular cytogenetic techniques is provided by Iannuzzi et al. [11].

Two examples of the importance of the use of FISH for the correct identification of the chromosomes involved in chromosome abnormalities of cattle were a case of autosome trisomy and two types of Robertsonian translocations. A case of autosome trisomy 28 in an abnormal calf, revealed by both R-banding and FISH mapping with a specific molecular marker [33], was identified, and the same abnormality was reported earlier as trisomy 22 using only the banding technique [86]. Two robs earlier reported as rob (4;8) [87] and rob (25;27) [88] in cattle were later corrected as rob (6;8) and rob (26;29), respectively, using C-, G-, and R-banding and FISH mapping with specific molecular markers and the use of HSA painting probes [28].

Table 1 shows that FISH mapping applications were used for the diagnosis of chromosome abnormalities in both metaphase (the majority) and interphase cells, the latter applied to lymphocyte nuclei (Figure 2), sperm (Figure 3), oocytes, and embryos.

Concerning the studies on meiotic preparations, those performed on the synaptonemal complexes (SCs), especially in spermatocytes, were particularly important for establishing the regularity of the pairing processes during the pachytene substage of meiotic prophase in animals carrying chromosome abnormalities (reviewed in [89]). Recent analyses of meiotic preparations have been performed using immune fluorescence approaches and have provided more detailed information on SCs [90,91,92]. Other studies have addressed the fragile sites in the chromosomes of domestic animals (reviewed by [93]), and limited studies have used CGH and SNP arrays to establish possible genomic losses occurring during chromosome rearrangements (Table 1).

FISH mapping was also very important for the definitive establishment of the agreement between various chromosome nomenclatures due to some discrepancies found during the Reading conference [94] and the subsequent ISCNDA1989 [95] (the inverted position between BTA4 and BTA6, as well as the correct position of BTA25, BTA27, and BTA29). This aspect was vital for the clinical cytogenetics of domestic bovids, as it allowed a correct identification of the chromosomes involved in chromosome abnormalities. During the Texas conference [96], specific molecular markers (only type I loci) were selected for each bovine syntenic group and each cattle chromosome based on previous standard chromosome nomenclatures. 

The next advance was the application of FISH mapping by two labs that used 31 selected BAC clones (from the Texas Conference) on RBG- and QBH-banded cattle preparations [97]. The chromosome-banding homologies among bovids (cattle, sheep, goats, and river buffalo) were then used to establish a definitive standard chromosome nomenclature for the main domestic bovid species [98]. Subsequent studies using FISH mapping and the same Texas markers on river buffalo, sheep, and goat R-banded chromosomes [99,100] definitively confirmed the chromosome homologies among domestic bovids, as established at the ISCNDB2000 [98].

### 2.2. FISH in Physical Mapping

The identification of the DNA structure [101] paved the way for the development of in situ hybridization technology. In the early stages of its development, this technology allowed the localization of genes using radioactive probes [102]. It was also used in studies of domestic animals [103,104], but the greatest diffusion of the physical mapping of genes awaited the development of fluorescent probes [105]. At that moment, we entered the golden years of gene mapping, and domestic animals were not excluded. One of the first examples was the localization of bovine alpha and beta interferon genes [106], and this localization was rapidly replicated in buffalos, goats, and sheep [107,108]. Subsequently, many other localizations were obtained using this technology (Figure 4). 

Considering the practical impossibility of compiling a complete list of all gene localizations obtained using this technology, some significant examples are listed in Table 2.

Localization sometimes involved a single gene [124,129] or a family of genes [132]. Other reports, however, mapped many genomic markers [100,141]. A point to remember is that FISH technology has significantly benefited from the availability of BAC genomic libraries—elements that represent the ideal source for the construction of the probes. Among these, the INRA library [144] and the CHORI-240 have played relevant roles. The publication of genomes [145,146,147,148] has since inevitably diminished interest in using this technology for mapping genetic factors, although genetic factor mapping continued for species whose genomes were sequenced later, such as the water buffalo [149]. However, this technology has proved useful in several aspects, including: a) the identification of errors in genomic assembly [150]; b) the refinement of genome assembly [151]; and c) the mapping of sequences not included in genomic assemblages [152]. Clearly, the interest today is very limited in locating a genetic factor in a species whose genomic sequence is available, but this does not mean that FISH technology is no longer indispensable for solving other problems related to the organization of genomes.

The mapping of genomic elements by FISH has also been used successfully for the physical mapping of data obtained by other technologies. The first examples concerned the physical anchoring of a genetic map to a chromosome [153,154,155] and the mapping of a synteny group to a specific chromosome [114]. Subsequent examples of the combined use of FISH and genetic maps followed [127,156].

### 2.3. Comparative FISH Mapping

Two main methods have been applied thus far to obtain a FISH mapping comparison between related and unrelated species: Zoo-FISH, which uses chromosome painting probes, and FISH mapping, which uses specific molecular markers of both type I and type II. Zoo-FISH is a molecular technique that provides an easier comparison between related and unrelated species from a macro point of view. The term was first reported by [157], based on earlier studies that used genomic chromosome painting probes, obtained by cell sorter chromosomes, to compare related species [158,159,160]. 

Zoo-FISH was first applied in domestic animals when human chromosome painting probes became commercially available. This approach demonstrated the conservation of several human chromosome segments in both domestic bovids (Table 3) and other domestic species (reviewed in [161]).

The use of human-chromosome painting probes allowed the identification of a substantial number of human chromosome segments (around 50) in bovid chromosomes [175,176,217,218,219]. Zoo-FISH has also been applied to correctly identify some chromosomes involved in the chromosome abnormalities shown in Table 1. The availability of specific painting probes obtained by both cell sorting and/or by the microdissection of specific chromosomes (or chromosome arms) from domestic animals extended these studies to investigations between related species (Table 3). For example, in cattle, Zoo-FISH was applied to study X-Y aneuploidy in sperm [55] and in oocytes [58] (Table 1). An interesting approach was demonstrated in two studies characterizing two cases of goat/sheep [220] and donkey/zebra [221] hybrids using multicolor FISH (M-FISH), starting from painting probes obtained from microdissected river buffalo chromosomes (or chromosome arms) and from flow-sorted donkey chromosomes, respectively. 

Chromosome painting probes allow the delineation of large, conserved chromosome regions between related and unrelated species, as reported above. The use of comparative FISH mapping using several chromosome markers to map a single type I or type II locus along the chromosomes allows a more accurate establishment of the gene order within chromosome regions, thereby confirming that chromosome rearrangements occurred to differentiate related or unrelated species in key evolutionary studies (Table 3). These detailed comparisons have confirmed a high degree of autosome (or chromosome arm) conservation among all bovid species. The main autosome difference found thus far in bovids was a chromosome translocation of a proximal chromosome region from *Bovinae* chromosome 9 to *Caprinae* chromosome 14, as demonstrated by both chromosome banding and, in particular, by a molecular marker (COL9A1) mapping proximal to *Bovinae* chromosome 9 and proximal to *Caprinae* chromosome 14 (reviewed in [9]). This translocation involved a genome region of about 13 MB and was followed by an inversion in *Caprinae* chromosome 14, as demonstrated earlier [213]. This chromosome event was common to all remaining *Bovidae* subfamilies, leading to the conclusion that the *Bovinae* subfamily is an ancestor to the remaining *Bovidae* subfamilies (reviewed in [9]). 

In contrast to autosomes, sex chromosomes are differentiated by more complex chromosome rearrangements. Indeed, the *Caprinae* X chromosome (as for all remaining X chromosomes of the other *Bovidae* subfamilies) is differentiated from the ancestor *Bovinae* X (very probably a large acrocentric chromosome, such as that of the water buffalo) by at least three chromosome transpositions and one inversion (reviewed in [9]). Detailed FISH mapping data are also useful for better anchoring of both genetic and RH maps [203,222,223,224]. The availability of detailed cytogenetic maps in bovid species allowed a better comparison of the bovid and human chromosomes, especially using type I loci. These comparisons facilitated the translation of genomic information from the human genome to the genomes of domestic animals, especially in those with no genome sequencing available. These comparisons also revealed a very high number of chromosome rearrangements that differentiate bovid species from humans. Indeed, the conservation of entire chromosomes or large regions of them between bovid and human chromosomes, as revealed by Zoo-FISH, was the result of complex chromosome rearrangements that differentiated human and bovid species according to their gene order. An example is presented in Figure 5 which illustrates the comparison of FISH mapping between HSA2q and BTA2. As seen, when utilizing the Zoo-FISH technique with the HSA2q painting probe, almost all BTA2 is painted [217], indicating a high degree of chromosome conservation between the chromosomes of the two species. By conducting the same comparison using comparative FISH mapping and examining the gene order along the chromosomes of the two species, we observe a distinct gene order between the two species, thus revealing complex chromosome rearrangements that differentiated the chromosomes of the two species during their evolution.

### 2.4. Fiber-FISH

The various FISH mapping techniques developed for human cytogenetics (reviewed by [225]) include SKY-FISH (spectral karyotyping FISH), Q-FISH (quantitative FISH), M-FISH (multicolor FISH), heterochromatin-M-FISH, COBRA-FISH (combined binary ratio labeling FISH), cenM-FISH (centromere-specific M-FISH), and fiber-FISH. Among these techniques, only fiber-FISH and M-FISH have been applied to domestic bovids. The use of fiber-FISH yields high-resolution maps of chromosomal regions and related genes on a single DNA fiber. This approach establishes the physical location of DNA probes with a resolution of 1000 bp. It is particularly useful for detecting gene duplications, gaps, and variations in the nuclear genome. The DNA fibers are obtained from nucleated cells by releasing the DNA fibers from the nucleus, stretching them mechanically, and then fixing them on slides [226] (Figure 6). Table 4 summarizes the studies that have used this technique in domestic bovids.

### 2.5. CGH Arrays

The CGH array technology, an evolution of in situ comparative genomic hybridization (CGH), is a method of cytogenetic investigation that emerged in the 1990s to overcome the limitations of common banding cytogenetic analyses, especially those involving the presence of genomic imbalances, such as duplications or deletions [231,232]. In situ CGH technology has many similarities to FISH: the support used is the same, i.e., denatured metaphases fixed on slides and the approaches to label the probes are identical. However, in this case, the probes are produced using complete genomic DNA deriving from two subjects: typically, one healthy and one relating to the subject being investigated. The two DNAs are labeled with two different fluorochromes and then hybridized simultaneously on the slide. In the hybridization phase, a competition is therefore created between the probes, and in the presence of a normal chromosomal segment, an intermediate color is obtained, while in the presence of chromosomal alterations, a fluorescence closer to one of the two colors used is obtained. Although this technology has been widely used and has provided important results, its major limitation lies in the resolution. CGH array technology follows the same principle, but the support is no longer represented by slides but by synthetic DNA fixed on slides. Initially, the chips for CGH array analyses contained DNA extracted from BAC to provide as uniform a representation of the genome as possible [233]. Current CGH array analyses are performed using devices containing oligonucleotides chosen that uniformly cover the whole genome and achieve resolutions of 5–10 kb [234,235]. More information about this technology and its use is provided by [236]. In species of zootechnical interest, CGH array analyses (Figure 7) became common following the appearance of the first commercial arrays, and these analyses are conducted essentially for two purposes: the identification of copy number variation (CNV) polymorphisms and the characterization of chromosome anomalies. CNVs are polymorphic variations present very frequently in the genomes of higher organisms [237,238,239]. In humans, approximately 4.8–9.7% of the genome contains CNVs [240]. The introduction of commercial arrays has allowed the use of this technology to obtain a great amount of information about the distribution of CNVs in species differences and how these variations are related to phenotypic traits. The transfer of this technology to the animal field and the availability of commercial arrays has led to the publication of several reports (Table 5).

## 3. Combined Informatic and Genomic Information

The publication of animal genomes [145,146,147,148,149,250] has made available a very large series of data that required the development of sophisticated analysis techniques and often required the use of computers with large processing capacities. The first bio-informatic analyses were used to assemble thousands of short genomic sequences, produced by modern high-throughput sequencing technologies, into genomes. Today, most of these programs are available free of charge through web pages that function as interfaces between the user and calculation tools [251]. Currently, dozens of bio-informatics programs are available to analyze the data contained in genomic assemblies, and many of these are accessible through various web platforms. Making a complete list is very complicated, in part because this is a rapidly evolving discipline that introduces, almost daily, new analytical tools.

### 3.1. Visualization of Genomes

The genomic sequences produced by the various assemblies can be visualized using one of the available websites available, including Genome Data viewer [252], UCSC Genome Browser [253], and Ensembl [254]. Currently, these websites provide the ability to view and process data relating to several genome assemblies (Table 6). 

These genome viewers are constantly evolving and contain several tools within them that allow the user to obtain highly relevant genetic data and information. This includes, but is not limited to, the possibility of: (a) identifying the structure of genetic factors (in terms of exon–intron boundaries); (b) identifying SNP polymorphisms in a particular region of the genome; (c) identifying the position of BACs by mapping the BES (Bac Ends Sequences, particularly useful when the user wants to choose the BACs to use in FISH analysis); (d) observing the genomic regions expressed in particular types of tissues; (e) analyzing the relationships between different assemblies of the same species; (f) visualizing the relationships between similar regions in different species (comparative genomics); and (g) viewing the repeating regions. In this review, we do not specify a best genome viewer, as this will often depend on personal needs and experience. However, as each genome viewer has its own specific analysis tools, sometimes the best solution is to use all three to obtain more complete information.

### 3.2. Use of Genomic Assemblies

The availability of genomic assemblages has, on the one hand, limited the interest in the physical mapping of genomic elements, but has, on the other hand, allowed the evolution of a very large number of genetic and genomic analyses. Probably one of the most common uses (even if not directly related to cytogenetics) is to design primers for use in PCR amplifications. This operation can be performed using different software, both available for free and for a fee. Among those available free of charge, the most frequently used is Primer3 [255]. The availability of genomic assemblages also makes rapid evolutionary investigation possible (i.e., visualizing, in a simple and rapid way, the similarities that exist between the various genomic regions of different species). The publication of genomes has certainly had a great impact on cytogenetics (both negatively and positively). If the golden era of gene mapping has ended, the possibility of rapidly identifying BACs for use as probes in FISH experiments has certainly provided great benefits to cytogenetics, as it avoids long and tedious testing of BAC libraries. This aspect has allowed the rapid characterization of some chromosomal anomalies, such as a centromere repositioning event in cattle [66], a reciprocal translocation, also in cattle [62], and cryptic evolutionary rearrangements between cattle and sheep [213]. Finally, the rapid localization of BACs on genomes has allowed the development of complex approaches for the identification of chromosomal abnormalities, which are also difficult to identify [71]. Obviously, these are not all the possible uses of genomic assemblies, but they represent the best examples in relation to cytogenetics. Each genomic assembly contains substantial information that can be used for very specific purposes and avoids the need for probes that would be complex to synthesize. The continuous evolution of these data analysis tools creates difficulty in any attempt to compile their possible uses.

### 3.3. Tools for Genomic Data Analyses

Simultaneously with the publication of the genomes, bio-informatics tools were developed for the analysis of the vast amount of data generated—data that are characterized by both their great variety and their large quantity. One of the main repositories of tools for analyzing genomic data is Galaxy [251]. This repository provides access to bio-informatic analysis tools, which are constantly updated. SNP variations represent the major source of variation in genomes, and the genomes of the species covered in this review are no exception. Currently, identifying these sources of variation is quite simple (through modern high-throughput sequencing techniques at ever-lower cost), but this does not characterize the effect that these variations can cause. For this scenario, the variant effect predictor (VEP available on the Ensembl website) software is helpful [256]. 

Without a doubt, BACs represent one of the most useful tools for molecular cytogenetics, and, as previously mentioned, their identification in genomes is currently greatly facilitated. However, the current situation would not be possible without the existence of two important institutions that have dedicated part of their activities to the construction, maintenance, and distribution of BAC libraries: the BACPAC Resources Center (BPRC, https://bacpacresources.org/ (accessed on 2 March 2023)) and INRA (http://abridge.inra.fr/index.php?option=com_flexicontent&view=item&cid=17&id=61&Itemid=202&lang=fr (accessed on 2 March 2023)). Through these two institutes, BACs belonging to different libraries can be obtained.

### 3.4. Whole-Genome Sequencing

In recent years, the decreasing costs of sequencing have made it possible to analyze many subjects. The purposes of these sequencings are different; in many cases, the aim is the identification of signatures of selection [257,258,259], but other purposes are represented, such as: (a) the identification of genetic variants in specific genes [260]; (b) the verification of data obtained regarding the identification of SNPs with chip arrays [261]; (c) the identification of the run of homozygosity in breeds intended for different productions [262]; (d) prediction and QTL mapping [263]; and (e) the identification of copy number variants [264] and transcriptome characterization [265]. Similar analyses were performed on sheep [266,267] and goats [268,269]. Additionally, in this case, the water buffalo seems to be slightly behind, as there are very few papers available on it [265].

## 4. PCR-Based Methods and Molecular Cytogenetics

The polymerase chain reaction (PCR) [270] is a method largely used to make millions of copies of a specific DNA sample in a fast and economical way for the detection, quantification, and typing of infectious diseases and genetic changes. Current PCR-based methods are distinguished as: (a) first-generation PCR, (b) second-generation quantitative PCR (qPCR), and (c) third-generation droplet-based digital PCR (dPCR). PCR detects endpoint, qualitative, or semi-quantitative assays by gel electrophoresis, separating DNA fragments according to size. The qPCR measures DNA/RNA in real time using PCR methods, fluorescent dyes, and fluorometry for relative quantification and quantitative assays with standard curves. The dPCR splits a PCR sample labeled with fluorescent dye into millions of microsamples to digitize the pool of DNA molecules with a single or no copy in each droplet. It quantifies the DNA/RNA copy number faster than qPCR based on standard curves [271]. 

In recent years, PCR-based methods have replaced the classic cytogenetic techniques for detecting chromosome abnormalities and aneuploidy due to greater precision, lower cost, and faster data than are possible with cytogenetic methods, because of the small quantities of DNA (30 ng) required from any stored or fresh biological samples. PCR-based approaches are most commonly used in bovid studies to examine sex chromosomes in early-sex-determination assays to detect aberrations (Table 7). 

Telomere assessment is another critical goal of cytogenetics research due to the central roles of telomeres in chromosome stability, aging, cancer development, apoptosis, and senescence. The telomeres consist of thousands of noncoding repetitive sequences of DNA composed of six nucleotide motifs (TTAGGG)n localized at the ends of chromosomes and are responsible for maintaining DNA integrity during each cell division. They are associated with several proteins, with the most abundant being the shelterin complex, which is made up of six different polypeptides. Telomeres also contain other genomic structures, such as T-loops, D-loops, G-quadruplexes (G4), R-loops, and long noncoding RNA (TERRA) [286]. 

In farm animals, telomere length (TL) did not receive much interest initially due to the difficulty in determining the natural limits of their lifespans. However, a recent study related TL to health, genome stability, and aging in cattle aged between 2 and 13 years and transformed TL into a sensitive biomarker for longevity and wellness (critical traits of selective breeding), responding to the “One Health” approach (improving animal welfare) [287]. TL is not often used as a unique marker of aging in humans because of its poor predictive accuracy due to increased telomere shortening in elderly humans as a consequence of age-related diseases (e.g., cancer, atherosclerosis, autoimmune disorders, obesity, chronic obstructive pulmonary disease, diabetes, hematological disorders, and neurodegenerative diseases) [288]. By contrast, TL proved to be a relevant biomarker of the general state of farm animals due to their lack of age-related pathologies [289,290].

Approaches for measuring TL include: (a) telomere restriction fragment (TRF) length [291]; (b) length analysis by Southern blotting; (c) fluorescent in situ hybridization (FISH) by flow cytometry (flow-FISH) or in metaphase cells (Q-FISH) [292,293]; and (d) PCR-based methods. Most of these methods have several limitations. For example, TRF and flow-FISH are labor-intensive and expensive; Southern blot analysis requires large amounts of genomic DNA, and Q-FISH works only on chromosomes (metaphase stage). Of the available methods, the PCR-based ones are the fastest, most recent, and least costly and require only small quantities of DNA (30 ng) from stored or fresh biological samples [294]. The qPCR method amplifies telomere repeats relative to a single-copy gene (reference gene) according to a method described by Cawthon et al. [295] and follows the MIQE guidelines [296]. One limitation of qPCR is the inconsistent repeatability and reproducibility of different TL measurement methods, producing a high variation in results [297]. Several studies on humans and animals indicated that the DNA extraction method might affect TL measurements using q-PCR, as DNA yields were higher using the non-silica membrane kit (salting-out method), and DNA integrity on electrophoresis gels varied [298,299]. A recent study showed comparable results for DNA quality and purity (tested using a NanoDrop instrument and electrophoresis gels) in cattle blood and milk samples using two different extraction kits (a salting-out kit for blood and a silica membrane kit for milk samples) due to the difficulty of extracting DNA from milk matrices. The DNA quality results were similar in both matrices, demonstrating a synchronous trend between them for the first time [287].

## 5. Current Developments and Knowledge Gaps

Molecular cytogenetics is approaching its first 30 years of history and during this period, it performed important functions that evolved over time. It therefore seems normal that in the coming years, we will witness further developments; however, some approaches will always be current and irreplaceable. The FISH technology represents, and will represent, the main methodology for the verification of chromosomal anomalies eventually identified with other approaches, just as the CGH array technology that will be increasingly used for the identification of genomic variants linked to a particular phenotype. Molecular cytogenetics could be very useful for the study of those species which have not yet benefited from the genomic revolution, or which are still in its early stages: in this sense, the water buffalo (Bubalus bubalis) is the main example. Despite possessing a great economic importance, its genome has been decrypted and made available only recently, and the application of other technologies is very late. A further gap that can be filled is the development of a technological approach that can allow the identification of all chromosomal types identifiable by cytogenetic analyses. A similar approach has already been published [71], but only the transfer of SKY-FISH technologies [300] from humans to bovids will bridge this gap. Finally, the certain decrease in costs will mean that even the species considered in this review will be able to benefit from long-read genomic sequencing, such as PacBio [301] and Oxford Nanopore [302].

## 6. Conclusions

The study of the chromosomes of domestic bovids is about to enter its seventh decade, and, as expected, it has undergone a notable evolution along the way. This evolutionary process for this discipline is mainly a result of the appearance of technologies that have significantly increased the potential of applied cytogenetics. Banding techniques, FISH, CGH arrays, and PCR have radically changed animal cytogenetics, making them irreplaceable tools for understanding the genetics of bred animals. Therefore, considering the history of cytogenetics, a quite easy prediction is that even the next evolutions will be dictated by technological advances. Predicting the next technological leap is difficult, but if we were to make a prediction, it would be that long-read genomic sequencing technologies will have important impacts on cytogenetics. Cytogenetics will likely retain its functionality, particularly in the confirmation of genomic results and the characterization of cytogenetic anomalies, as well as in evolutionary studies. This is because the most significant genetic mutations have accumulated at the chromosome level during the evolution of species. Finally, the implication and progresses from animal cytogenetics can be summarized as follows:In the pre-genomic era, FISH technology represented the almost exclusive technology available for the localization of genes in genomes.Prior to the availability of low-cost genomic sequencing, molecular cytogenetics was the only approach for identifying similarities between karyotypes of different species.The technologies of molecular cytogenetics represent the best approach for the characterization of chromosomal abnormalities.Despite scientific progress in similar disciplines, molecular cytogenetics will always find its place and represent an inescapable investigation methodology.

## Figures and Tables

**Figure 1 animals-13-00944-f001:**
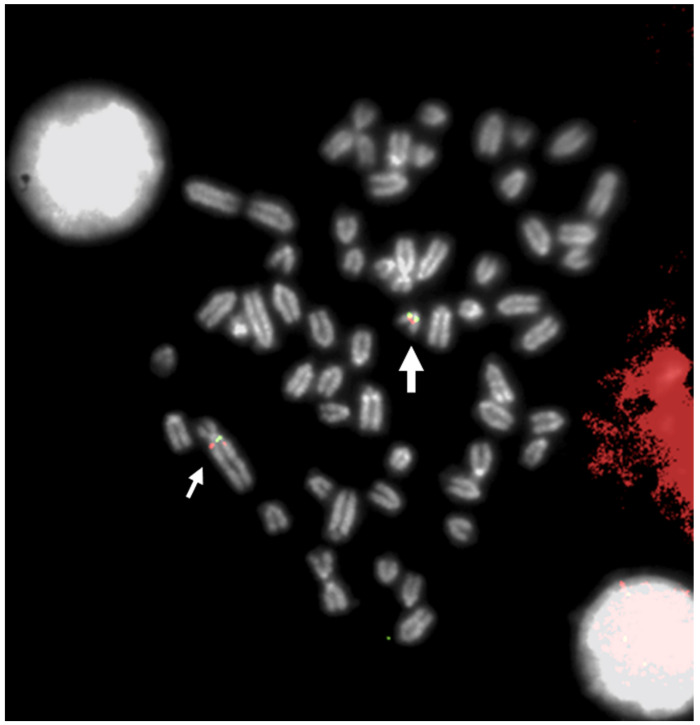
FISH mapping with a BAC clone mapping proximal to BTA29 (large arrow) and proximal to q-arms (BTA1) of rob (1;29) (small arrows). Indeed, a small chromosome region of 5,4 Mb translocated from proximal BTA29 to the proximal region of BTA1 (with an inversion), originating rob (1;29) [56]. Different colors indicate different BACs.

**Figure 2 animals-13-00944-f002:**
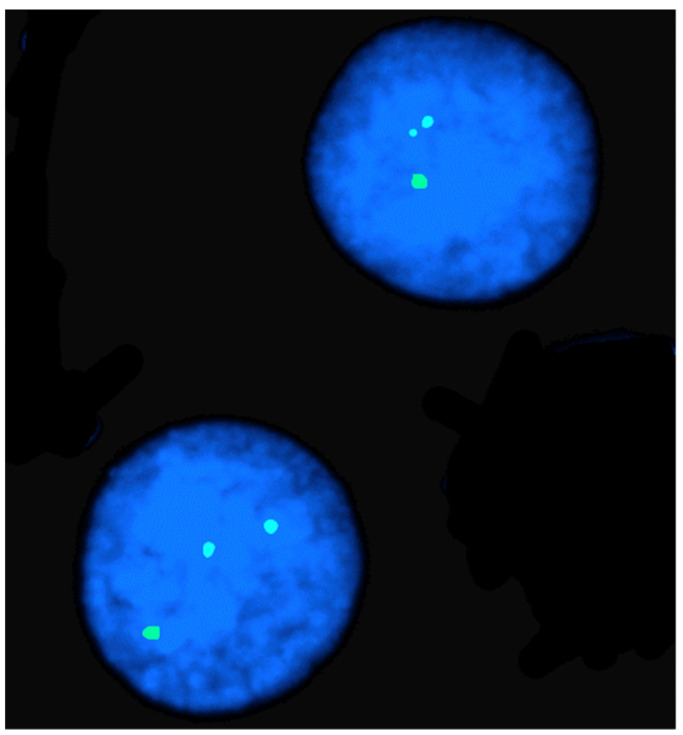
FISH mapping in an interphase nucleus of a female river buffalo affected by X-trisomy. Note the three hybridization signals due to the X chromosome PGK marker.

**Figure 3 animals-13-00944-f003:**
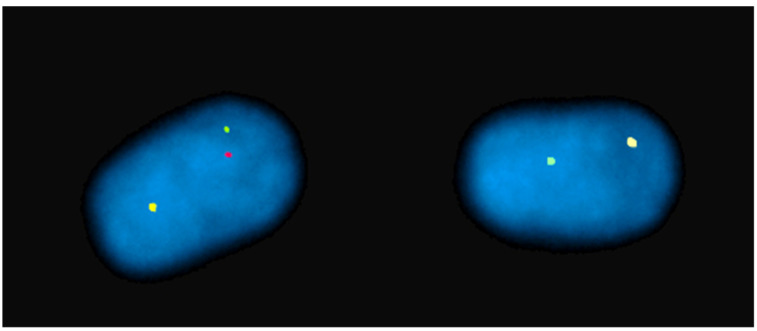
Sperm-FISH in a river buffalo bull carrying a rob (1p;18) using BAC probes for BBU 1p (red), BBU 1q (green), and BBU 18q (yellow) chromosomes. Normal sperm nucleus with 1/1/1 fluorescent phenotype and separate signals on left. Unbalanced sperm nucleus with 1/0/1 fluorescent phenotype on right.

**Figure 4 animals-13-00944-f004:**
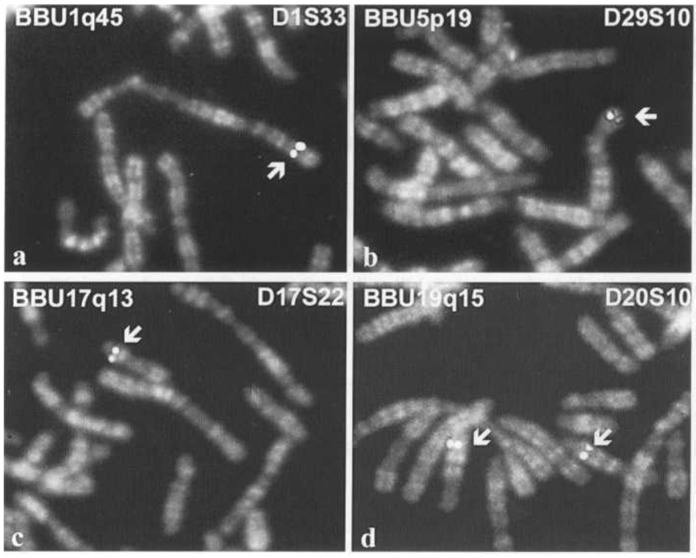
FISH mapping of type II loci in river buffalo R-banded chromosomes. FITC signals (arrows) of the markers and RBH banding were separately acquired by two different microscope filter combinations. Then signals were precisely superimposed to R-banded chromosomes (*Drawn from Iannuzzi et al., Cytogenet Cell Genet. 102, 65–75, 2003, DOI: 1 0.1159/000075727, S. Karger AG, Basel* [109]).

**Figure 5 animals-13-00944-f005:**
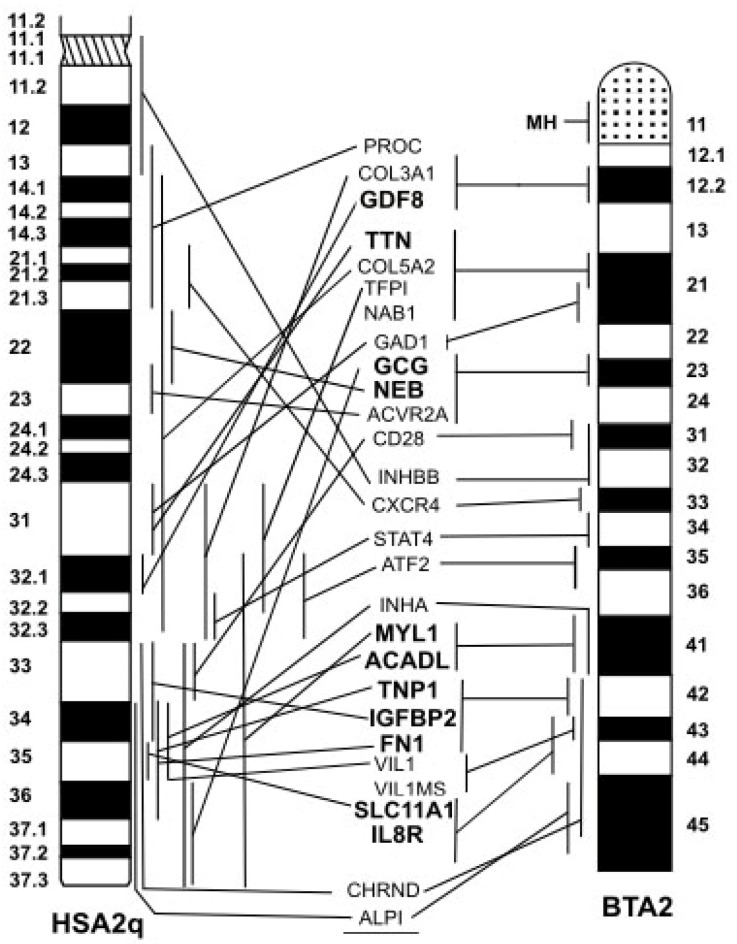
Comparative FISH mapping between HSA2q and BTA2. Note the different gene order between the two chromosomes due to complex chromosome rearrangements occurred during the chromosome evolution of the two species (*Drawn from Di Meo et al., Animal Genetics 37, 299–300, 2006, Wiley Online Library* [140]).

**Figure 6 animals-13-00944-f006:**
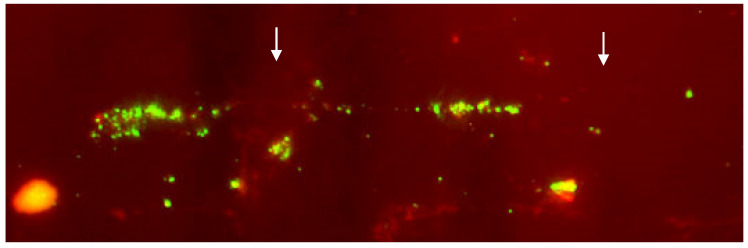
Details of the fiber-FISH performed on a lymphocyte nucleus of cattle affected by arthrogryposis using a BAC clone containing the survival of motor neuron gene (SMN). The presence of two groups of linear hybridization signals (arrows) supports the hypothesis that SMN was at least duplicated [135].

**Figure 7 animals-13-00944-f007:**
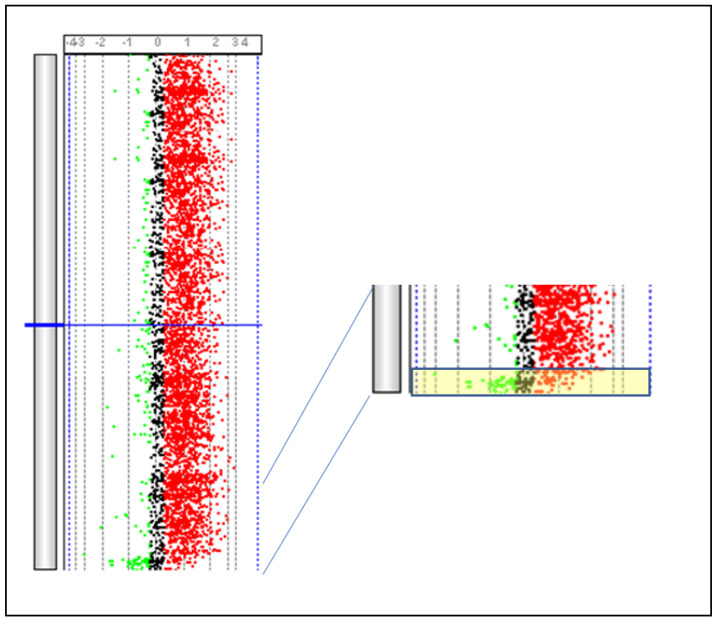
Identification of the PAR region present on BTAX and BTAY. The PAR region (yellow box) is identified by comparing DNA obtained from a male subject and that obtained from a female subject using a SurePrint G3 Bovine CGH Microarray 180 k (Agilent Technologies, Santa Clara, CA, USA). *Parma P. Personal communication*.

**Table 1 animals-13-00944-t001:** FISH mapping approaches applied for the detection of chromosome abnormalities in domestic bovids. The type of chromosome abnormality, the techniques used (including FISH), the main results, and authors are reported.

Species	Chromosome Abnormality	Techniques Used	Main Results	References
Cattle	t(X-BTA23) in two normal cows	QBH, FISH	Better position of MHC-locus	[22]
	Minute fragment	Bovine SAT-DNA	Visualization of fragment	[23]
	rob(4;10)	Bovine bivariate flow painting probes on R-banded karyotype	Discovery of a new rob	[24]
	iso(Yp)	GTG, FISH with repeat sequences	Visualization of iso(Yp)	[25]
	Trisomy 20	QBH, FISH	Malformed calf with cranial defects	[26]
	rob(2;28)	Q-, R-banding, telomeric probe	Monocentric translocation	[27]
	rob(1;29), rob(6;8), rob(26;29)	GBG, RBG, CBA, FISH, HAS painting probe	correct identification of two of the three robs earlier published	[28]
	Mixoploidy	Dual-color FISH with BTA6/BTA7 painting probes	72% of IVP blastocysts were mixoploid, versus 25% in vivo	[29]
	Mixoploidy/polyploidy	Dual-color FISH with BTA6 and 7 painting probes on in vitro embryo cells	Numerical chromosome aberrations were detected as early as day 2 post insemination (pi)	[30]
	rcp(1;5)(q21;qter)(q11;q33)	CBA, GBG, RBG, FISH with HSA3 and HSA12 painting probes	Bull and dam carriers, the latter with poor fertility	[31]
	invY(Yq11-q12.2)	CBG, RBA, FISH	12 young males of which one (carrier) had female traits	[32]
	Trisomy 28	CBG, RBA, FISH	New chrom. identification of a previous studied case of abnormal calf	[33]
	t(Xp+;23q-)	FISH with painting probe, SCA	Oligospermic bull	[34]
	rcp(Y;9)(q12.3:q21.1).	CBA, RBG, FISH	Azoospermic bull	[35]
	Polyploidy	Painting probe BTA6 and BTA7 by microdissection on in vitro embryos	Polyploidy was significantly higher in trophectoderm (TE) cells than in embryonic disc (ED) cells	[36]
	rob(1;29)	FISH with SAT-I, III, IV	Different pattern of satellite DNA families in several chromosomes, model of rob(1; 29) origin	[37]
	Mosaicism 2 n = 60/2 n = 60 t (2q−;5p+)	FISH with painting probes BTA2 and BTA5	Translocation mosaicism in a bull	[38]
	XXY-Trisomy	X-Y painting probes	Testicular hypoplasia	[39]
	fragm/hypoploidy/hypoploidy-mixoploidy; hyperploidy/hyperploidy-mixoploidy	Karyotyping, FISH with X-Y painting probes in nuclear transfer embryos	Anomalies occurred in NT embryos varied according to the donor cell culture and paralleled the frequency of anomalies in donor cells	[40]
	rob(1;29)	CBG, GTG, FISH with a rob(1;29) painting probe	Presence of rob(1;29) in Gaur (*Bos gaurus*)	[41]
	rob(1;29)	CBA, RBA, FISH	Origin of rob(1;29) by complex chromosome rearrangements	[42]
	rob(1;29)	Sperm-FISH	Low percentage of abnormal sperm in two carriers	[43]
	rcp(9;11)(q27;q11)	RBG and FISH	De novo origin of the rcp	[44]
	Mosaicism XX/XY cells	FISH with a male-specific BC1.2 DNA sequence in interphase cell nuclei	Diagnosis of freemartin	[45]
	rcp(11;21)(q28-q12)	CBA, RBA, Ag-NORs, FISH	Normal bull but with absence of libido; reduced fertility (very low presence of spermatozoa in germinal elements)	[46]
	rob(1;29)	microdissection, DOP-PCR, cloning and sequencing, sperm-FISH	Detection of sperm-carrying rob(1;29)	[47]
	rcp(2;4)(q45;q34)	G-banding, SCA, and chromosome painting	Detection of a new rcp in bull	[48]
	Aneuploidy	Dual-color FISH with Xcen/Y painting probes in sperm	Study the aneuploidy in different breeds	[49]
	rcp(4;7)	RBG, FISH (painting probe), aCGH	Normal male and no genomic loss in the rcp	[50]
	Aneuploidy	Dual-color FISH with Xcen and BTA5 painting probes	Study of aneuploidy in oocytes of two breeds	[51]
	Aneuploidy	FISH with BTAX, BTAY, and BTA6 painting probes on sperm of several young bulls	Aneuploidy frequencies in young fertile bull spermatozoa were relatively low	[52]
	rcp(Y;21)(p11;q11)	G-banding, FISH	Normal young bull but lower testosterone level at 12 months	[53]
	rcp(11;25)(q11, q14∼21)	CBA, RBA, FISH, NOR	der11 with two C-bands for a break at the centromere of BTA25; cow with reduced fertility	[54]
	Aberrant oocytes	Dual-color FISH of X-cent/BTA5 painting probes	Similar rate of aneuploidy in different cattle breeds	[55]
	rob(1;29)	FISH, aCGH	New results of the origin of this rob by transposition, inversion; no gene-coding regions were disrupted during the rearrangements	[56]
	Xp-del (inactive X)	CBA, RBA, FISH	del found in both dam and calf (normal cow)	[57]
	X-Y aneuploidy	Dual-color FISH with Xcen-BTAY painting probes	Testing X-Y ratio and aneuploidy	[58]
	Aneuploidy	Dual-color FISH with Xcen and five autosome painting probes	Similar rates of chromosomal aberrant secondary oocytes in two indigenous cattle breeds	[59]
	Mixoploidy	FISH with BTAX and BTA6 painting probes	First zygotic cleavage (FZC) is a marker of embryo quality by demonstrating a significantly lower incidence of aberrations in early embryos	[60]
	Aneuploidy/polyploidy	CA, SCE, MN, MI, FISH	Effect of the tebuconazole-based fungicide: monosomies and trisomies on BTA5 and 7	[61]
	rcp(5;6)(q13;q34)	RBG, FISH, aCGH	Normal young bull with balanced rcp	[62]
	rcp(13;26)(q24;q11)	CBG, GTG, painting probes BTA13 and 26, telomeric probe	De novo rcp in both dam and calf	[63]
	der(11)t(11;25)(q11;q14–21)	CBA, RBA, FISH	Abnormal female calf	[64]
	Chromosome damages	SCE, MN, FISH with BTA1, 5, 7 painting probes	No significant chromosome fragility with use of thiacloprid	[65]
	Abnormal BTA17 in a young bull	CBA, R-banding, FISH, PNA-telomeric probe, aCGH, SNP array	Centromere repositioning	[66]
	X-monosomy	Karyotyping, FISH, SNP genotype data	Sterile for abnormal internal sex adducts	[67]
	rob(3;16)	Sperm-FISH	Low rate of unbalanced gametes produced by adjacent segregation (5.87%) and interchromosomal effect (ICE) on BTA17 and BTA20	[68]
	Trisomy 20	Q-banding, FISH	Malformed fetus, cranial defects	[69]
	Trisomy 29	FISH/genomic analysis	Malformed female calf showing dwarfism with severe facial anomalies	[70]
	rob(1;29); rcp(12;23)	FISH, use of BAC clones mapping prox- and dist- regions of all cattle autosomes and X	Identification of chromosome abnormalities in all autosomes and BTAX	[71]
	tan(18;27)	CBA, RBA, FISH	Male calf with congenital hypospadias and a ventricular septal defect	[72]
River buffalo	X-monosomy	CBA, RBA, FISH	Normal body conformation and external genitalia, ovaries not detectable, sterile	[73]
	rob(1p;23)	CBA, RBA, Ag-NORS, FISH	Complex chromosome abnormality with fission on BBU1 and centric fusion of BBU1p with BBU23 in both dam and female calf; reduced fertility in the dam	[74]
	rob(1p;18)	CBA, RBA, FISH	Famous bull eliminated from reproduction for the presence of the same chrom. abnormality in part of progeny	[75]
	Chromosome abnormalities	Zoo-FISH	Sequential approach with 13 chromosome river buffalo painting probes to detect river buffalo chromosome abnormalities	[76]
	rob(1p;18)	Sperm-FISH in motile and total fraction sperm	Limited effects on the aneuploidy in gametes on the motile fraction sperm	[77]
River/Swamp buffalo	Aneuploidy	M-FISH	Study of aneuploidy in river and swamp buffalo oocytes	[78]
Sheep	Chromosome abnormality	Production of all sheep chromosome painting probes from cell sorter technique	Easy identification of chromosome abnormalities	[79]
	rob(8;11)	G-bands, painting probes 8 and 11, SAT-I and SAT-II	SAT-I proximal on both arms with SAT-II covering the centromere	[80]
	Diploid-polyploid mosaicism	Zoo-FISH with bovine painting probes X/Y and 1;29 on nuclei of in vivo and in vitro embryos	In vitro embryos showed significant higher number of abnormal embryos than in vivo ones	[81]
	del(10q22)	Use of ovine BAC clone in addition to genetic analyses	Micro-chromosomal deletion responsible for EDNRB gene lack	[82]
	rcp(4q;12q)(q13;q25)	CBA, RBA, FISH with both specific markers and PNA-telomeric probe	Characterization of a new rcp in a young sheep	[83]
	rcp(18;23)(q14;q26).	CBA, RBA, FISH with bovine painting probe	Reduced fertility	[84]
	Chromosome abnormalities in bovids	Partial river buffalo chromosome painting probes from microdissection	Detection of chromosome abnormalities in bovids	[85]

**Table 2 animals-13-00944-t002:** Gene mapping obtained with FISH in domestic bovids. Type I and type II markers are expressed with polymorphic (SSRs, microsatellite, STSs) sequences, respectively.

Gene/Genes/Marker	Species	Reference
Lysozyme gene cluster	BBU	[110]
Uridine monophosphate synthase	BTA	[111]
Uridine monophosphate synthase	BBU	[112]
BTA1 to 7	BTA	[113]
Microsatellites	BTA	[114]
Microsatellites	BTA	[115]
Beta-defensin genes	BTA; OAR	[116]
Alpha-S2 casein	BTA; BBU	[117]
Fas/APO-1	BTA	[118]
Interferon gamma	OAR	[119]
Interleukin-2 receptor gamma	BTA	[120]
Beta-lactoglobulin pseudogene	BTA, OAR, CHI	[121]
Bone morphogenetic protein 1	BTA	[122]
TSPY	BTA, OAR, CHI	[123]
VIL	OAR, CHI, BBU	[124]
Type I markers	BTA	[125]
Prion protein gene	BTA, OAR, CHI, BBU	[126]
IL2RA, VIM, THBD, PLC-II, CSNK2A1, TOP1	BTA	[127]
NF1, CRYB1, CHRNB1, TP53, P4HB, GH1	OAR, BBU	[128]
PAX8	BTA, OAR, CHI	[129]
Type I markers	BTA	[97]
PREF1	BTA	[130]
PRKCI	BTA	[131]
MHC	BTA	[132]
Type I markers	OAR, CHI	[100]
CACNA2D1	BTA	[133]
SLC26a2	BTA	[134]
SMN	BTA, OAR, CHI, BBU	[135]
Type I markers	BBU	[109]
Type I and II markers	OAR	[136]
PRPH	BTA	[137]
CYP11b/CYHR1	BTA	[138]
SRY, ANT3, CSF2RA	BTA	[139]
Autosomal loci (11)	BTA, OAR, CHI, BBU	[140]
Autosomal loci (88)	OAR	[141]
Autosomal loci (68)	BBU	[142]
BMPR1B, BMP15, GDF9	BTA, OAR, CHI, BBU	[143]

**Table 3 animals-13-00944-t003:** Comparative FISH mapping in domestic bovids with related and unrelated species.

Author/s	Results
[107]	Mapping omega and trophoblast interferon genes in cattle and river buffalo
[162]	Mapping of lactoperoxidase, retinoblastoma, and alpha-lactalbumin genes in cattle, sheep, and goats
[108]	Mapping omega and trophoblast interferon genes in sheep and goats
[163]	Mapping LGB and IGHML in cattle, sheep, and goats
[164]	Mapping CASAS2 gene to the cattle, sheep, and goat chromosome 4
[165]	Mapping MHC-complex in cattle and river buffalo
[166]	Mapping inhibin-alpha (INHA) to OAR2 and BTA2
[167]	Mapping inhibin subunit beta b to OAR2 and BTA2
[121]	Mapping beta-lactoglobulin pseudogene in sheep, goats, and cattle
[168]	Mapping ZNF164, ZNF146, GGTA1, SOX2, PRLR, and EEF2 in bovids
[117]	Mapping of the alpha-S2 casein gene on river buffalo and cattle
[116]	Mapping of beta-defensin genes to river buffalo and sheep chromosomes suggest a chromosome discrepancy in cattle standard karyotypes
[169]	Mapping STAT5A gene maps to BTA19, CHI19, and ORA11
[170]	Mapping in Y chromosomes of cattle and zebu by microdissected painting probes
[124]	Mapping of villin (VIL) gene in river buffalo, sheep, and goats
[126]	Mapping prion protein gene (PRNP) on cattle, river buffalo, sheep, and goats
[171]	Mapping BCAT2 gene to cattle, sheep, and goats
[172]	Comparative mapping in X chromosomes of bovids
[173]	Comparative mapping between BTA-X and CHI-X
[174]	Survey of chromosome rearrangements between ruminants and humans
[175]	Comparative mapping between cattle and pig chromosomes using pig painting probes
[176]	Extensive conservation of human chromosome regions in euchromatic regions of river buffalo chromosomes
[128]	Mapping of six expressed gene loci (NF1, CRYB1, CHRNB1, TP53, P4HB, and GH1) to river buffalo and sheep chromosomes
[177]	Comparison of human and sheep chromosomes using human chromosome painting probes
[178]	Mapping four HSA2 type I loci in river buffalo chromosomes 2q and 12
[179]	Mapping BCAT1 in cattle, sheep, and goats
[180]	Comparative mapping in bovid X chromosomes reveals homologies and divergences between the subfamilies *Bovinae* and *Caprinae*
[181]	Mapping 16 type I loci in river buffalo and sheep
[182]	Mapping 13 type I loci from HSA4q, HSA6p, HSA7q, and HSA12q on in river buffalo
[183]	Mapping forty autosomal type I loci in river buffalo and sheep chromosomes and assignment from sixteen human chromosomes
[184]	Mapping eight genes from HSA11 to bovine chromosomes 15 and 29
[98]	International chromosome nomenclature in domestic bovids based on Q-, G-, and R-banding and FISH with 31 specific Texas marker chromosomes
[185]	Mapping 28 loci in river buffalo and sheep chromosomes
[186]	Sheep/human comparative map in a chromosome region involved in scrapie incubation time shows multiple breakpoints between human chromosomes 14 and 15 and sheep chromosomes 7 and 18
[135]	Physical map of the survival of motor neuron gene (SMN) in domestic bovids
[100]	Assignment of the 31 type I Texas bovine markers in sheep and goat chromosomes by comparative FISH mapping and R-banding
[187]	Mapping 195 genes in cattle and updated comparative map with humans, mice, rats, and pigs
[188]	Mapping of F9, HPRT, and XIST in BTAX and HSAX clarifies breakpoints between the two species
[189]	15 gene loci were mapped in the telomeric region of BTA18q and HSA19q
[190]	Comparative G- and Q-banding of saola and cattle chromosomes as well as FISH mapping of 32 type I Texas markers
[191]	Mapping of fragile histidine triad (FHIT) gene in bovids
[192]	Chromosome evolution and improved cytogenetic maps of the Y chromosome in cattle, zebu, river buffalo, sheep, and goats
[193]	Physical map of mucin 1, transmembrane (MUC1) among cattle, river buffalo, sheep, and goat chromosomes and comparison with HSA1
[194]	Mapping of LEP and SLC26A2 in *bovidae* chrom. 4 (BTA4/OAR4/CHI4) and HSA7
[140]	Mapping 11 genes to BTA2, BBU2q, OAR2q, and CHI2, and comparison with HSA2q
[195]	Mapping among humans, cattle, and mice suggests a role for repeat sequences in mammalian genome evolution
[196]	Mapping sheep and goat BAC clones identifies the transcriptional orientation of T cell receptor gamma genes on chromosome 4 in bovids
[197]	Mapping of twelve loci in river buffalo and sheep chromosomes: comparison with HSA8p and HSA4q
[198]	Mapping 25 new loci in BTA27 and comparison with both human and mouse chromosomes
[141]	An advanced sheep cytogenetic map and assignment of 88 new autosomal loci
[199]	Cross-species FISH with cattle whole-chromosome paints and satellite DNA I probes was used to identify the chromosomes involved in the translocations of some tribe *Bovinae* species
[142]	Extended river buffalo cytogenetic map, assignment of 68 autosomal loci and comparison with human chromosomes
[200]	FISH with 28S and telomeric probes in 17 bovid species. NORs are an important and frequently overlooked source of additional phylogenetic information within the *Bovidae*
[201]	Mapping DMRT1 genes to BTA8 and HSA9
[202]	Comparative DM domain genes between cattle and pigs
[203]	Assignments of new loci to BBU7 and OAR6 and comparison with HSA4
[204]	Mapping 22 ovine BAC clones in sheep, cattle, and human X chromosome
[205]	Mapping and genomic annotation of bovine oncosuppressor gene in domestic bovids
[206]	Cytogenetic map in sheep as anchor of genomic maps also using different genomic resources from other species
[207]	Molecular cytogenetics in goats and comparative mapping with human maps
[208]	Mapping of 6 loci containing genes involved in the dioxin metabolism of domestic bovids
[209]	Extended cytogenetic maps of sheep chromosome 1 and their cattle and river buffalo homologues: comparison with the OAR1 RH-map and HSA2, 3, 21, and 1q
[210]	Mapping between BTA5 and some *Antilopinae* species using Sat-I and SAT-II sequence and BTA-painting probes
[211]	Comparison of centromeric repeats between cattle and other *Bovidae* species
[212]	Advanced comparative map in X chromosome of *Bovidae*
[143]	Physical map of BMPR1B, BMP15, and GDF9 fecundity genes on cattle, river buffalo, sheep, and goat chromosomes
[152]	Physical mapping of 20 unmapped fragments in Btau 4.0 Genome Assembly in cattle, sheep, and river buffalo
[213]	Physical map of LCA5L gene in cattle, sheep, and goats
[214]	New cryptic difference between cattle and goat karyotypes
[215]	Small evolutionary rearrangement between BTA21 and homologous OAR18
[216]	Assignment of 23 endogenous retrovirus to both sheep and homologous chromosomes regions of river buffalo

**Table 4 animals-13-00944-t004:** Studies using the fiber-FISH on domestic bovids.

Species	Author/s	Results
Cattle	[227]	Genomic organization of the bovine aromatase
	[228]	Molecular characterization of STAT5A- and STAT5B-encoding genes
	[135]	Demonstration of survival of motor neuron gene (SMN) duplication in a calf affected by arthrogryposis
	[229]	Demonstration of multiple TSPY copies on the Y chromosome
Sheep	[230]	DNA fiber barcodes indicated a chromosomal deletion

**Table 5 animals-13-00944-t005:** Identification of CNV.

Specie	Reference	Note
Cattle	[241]	3 Holstein bulls
Cattle	[242]	90 animals: 11 Bos taurus breeds, 3 Bos indicus breeds, and 3 composite breeds for beef, dairy, or dual purpose
Cattle	[243]	20 animals: 14 Holsteins, 3 Simmental 2 Red Danish and 1 Hereford
Cattle	[244]	47 Holstein bulls
Cattle	[245]	24 animals from Chianese breeds
Cattle	[246]	3 Angus, 6 Brahman, and 1 composite animal
Sheep	[247]	36 animals
Sheep	[248]	12 animals
Goat	[249]	10 animals

**Table 6 animals-13-00944-t006:** Independent genomic assemblies that can be analyzed through the main genomic visualization sites.

Specie ^1^	Genome Assembly ^2^	Origin	GDW ^3^	UCSC ^4^	ENS ^5^
BTA	ARS-UCD1.3	USDA ARS	yes	no	no
	ARS-UCD1.2	USDA ARS	no	yes	no
	Btau_5.0.1	Cattle Gen. Seq. Int. Consortium	yes	no	no
	Btau_4.6.1	Cattle Gen. Seq. Int. Consortium	no	yes	no
	Btau_4.0	Cattle Gen. Seq. Int. Consortium	no	no	yes
	UMD_3.1.1	University of Maryland	yes	yes	no
	UMD_3.1	University of Maryland	no	no	yes
	Baylor 4.0	Baylor College of Medicine	no	yes	no
OAR	ARS-UI_Ramb_v2.0	University of Idaho	yes	no	no
	Oar_rambouillet_v1.0	Baylor College of Medicine	yes	no	yes
	Oar_v4.0	Int. Sheep Gen. Consortium	yes	yes	no
	CAU_O.aries_1.0	China Agricultural University	yes	no	no
CHI	ARS1.2	USDA ARS	yes	no	no
	ARS1	USDA ARS	no	no	yes
	CHIR_1.0	Int. Goat Gen. Consortium	yes	no	no
BBU	NDDB_SH_1	Nat. Dairy Dev. Board, India	yes	no	no
	UOA_WB_1	University of Adelaide	yes	no	no
BIN	Bos_indicus_1.0	Genoa Biotecnologia SA	yes	no	no

^1^ BTA = cattle; OAR = sheep; CHI = goat; BBU = water buffalo and BIN = Zebu. ^2^ Only genomic assemblages at the chromosomal level were considered and not those limited to scaffolds. ^3^ Genome data viewer. ^4^ USCS genome browser. ^5^ Ensembl genome browser.

**Table 7 animals-13-00944-t007:** PCR-based approaches on bovids for the detection of chromosomal aberrations.

Species	Objective	Sample	PCR-Based Method	Reference
Cattle	Sex-determination	Embryos	PCR	[272]
Cattle	Freemartinism diagnosis	Blood	PCR	[273]
Cattle	Sex-determination	Embryos	PCR	[274]
Cattle	Sex-determination	Spermatozoa	PCR	[275]
Cattle	Chimerism diagnosis	Blood	qPCR	[276]
Cattle	XX/XY chimerism diagnosis	Blood	PCR	[277]
Cattle	SRY-positive hermaphrodite diagnosis	Blood	PCR	[278]
Cattle	XY (SRY-positive) diagnosis	Blood	PCR	[279]
Cattle	Freemartinism diagnosis	Blood	qPCR	[280]
Cattle	Freemartinism diagnosis	Blood	dPCR	[281]
Cattle	Sex-determination	Spermatozoa	dPCR	[282]
Cattle	Mosaic karyotype (60,XX/60,XX,+mar) diagnosis	Skin tissue	PCR	[283]
Cattle	Mosaicism (60,XX/90,XXY) diagnosis	Blood, skin, buccal epithelial cells, and hair follicles	dPCR	[284]
Cattle	XX/XY chimerism diagnosis	Blood and hair follicles	dPCR	[285]

## Data Availability

Data sharing is not applicable to this article as no new data were created or analyzed in this study.

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
