# Peer review of "Molecular Cytogenetics in Domestic Bovids: A Review"

_animals, 2023, doi:10.3390/ani13050944_

Round 1
Reviewer 1 Report
This manuscript presents a valuable extension of a recent review articles of the same authors, published also in ANIMALS journal [Iannuzzi, A., Parma, P., & Iannuzzi, L., 2021. Chromosome Abnormalities and Fertility in Domestic Bovids: A Review. Animals 11(3), 802].
There are several points, which are suggested to be considered to improve quality of the manuscript:
1. There are too many repetitions in „simple summary”, “summary” and “introduction”.
2. Figure 1 – could be deleted (similar figures are commonly presented in textbooks)
3. Table 1.
a) Title should be re-written, e.g.: FISH-mapping approaches applied for detections of chromosome abnormalities in domestic bovids.
b) Abrreviations should be explained in the footnote (e.g. GBG, RBG, RBA, CBA, QBH, IVP, SCA, CA, MN, MI. PNA, M-FISH
c) Last column – replace “Authors” by “References”
4. Table 2 and lines 175 and 183 . The terms “type I and II markers” and “Texas markers” should be explained
5. Line 243. The following reference should be cited: [Solinas-Toldo, S., Lengauer, C., & Fries, R. (1995). Comparative genome map of human and cattle. Genomics, 27(3), 489–496.]
6. Table 2 (second column) – should be “Species”
7. Figure 3 – should be” interphase” (not inter-phase)
8. Lines 262-267: two terms are used “transposition” and “translocation”. I suggest to use “translocation”. Was it reciprocal translocation?
9. Line 279: it is suggested to use “translation:" instead of “transfer”
10. Line 442 and table 7 – a correct term and the abbreviation is: droplet digital PCR (ddPCR)
11. Line 460: please add that the repetitive sequence is composed of six nucleotide motif (TTAGGG)n
12. Paragraph “2.5. CGH arrays”. It is suggested to explain how a classical CGH works.
Finally, since I am not a native speaker I suggest to ask for a proof reading of the manuscript by a native speaker.
Author Response
There are several points, which are suggested to be considered to improve quality of the manuscript:
- There are too many repetitions in „simple summary”, “summary” and “introduction”.
Following what is indicated, we have made several corrections.
- Figure 1 – could be deleted (similar figures are commonly presented in textbooks)
We have eliminated figure 1
- Table 1.
- a) Title should be re-written, e.g.: FISH-mapping approaches applied for detections of chromosome abnormalities in domestic bovids.
The title has been changed as suggested.
- b) Abbreviations should be explained in the footnote (e.g. GBG, RBG, RBA, CBA, QBH, IVP, SCA, CA, MN, MI. PNA, M-FISH
We have included the explanation of these acronyms at the end of MS for all ones, except for M-FISH reported in the Fiber-FISH section.
- c) Last column – replace “Authors” by “References”
We have made this change.
- Table 2 and lines 175 and 183. The terms “type I and II markers” and “Texas markers” should be explained.
We have made this change.
- Line 243. The following reference should be cited: [Solinas-Toldo, S., Lengauer, C., & Fries, R. (1995). Comparative genome map of human and cattle. Genomics, 27(3), 489–496.]
We have added this reference.
- Table 2 (second column) – should be “Species”
We have made this change.
- Figure 3 – should be” interphase” (not inter-phase)
We have made this change.
- Lines 262-267: two terms are used “transposition” and “translocation”. I suggest to use “translocation”. Was it reciprocal translocation?
We have now used the exact word: translocation.
- Line 279: it is suggested to use “translation:" instead of “transfer”
We have now used the exact word: translation.
- Line 442 and table 7 – a correct term and the abbreviation is: droplet digital PCR (ddPCR).
We have preferred using the more generic term dPCR (digital PCR technology) referring to a method that portions the PCR solution into tens of thousands of nano-liter sized. The term you suggested (ddPCR) is also correct but refers to a specific digital PCR based on water-oil emulsion droplet technology.
- Line 460: please add that the repetitive sequence is composed of six nucleotide motif (TTAGGG)
We have added this information.
- Paragraph “2.5. CGH arrays”. It is suggested to explain how a classical CGH works.
We have added a small paragraph on how this technology works.
- Finally, since I am not a native speaker I suggest to ask for a proof reading of the manuscript by a native speaker.
Before submitting the article, we had an English language proofread by an external proofreading company (Scribendi). We have been using this service for several years for our work and we have never had any problems.
Reviewer 2 Report
The review provides a very solid overview of molecular cytogenetics of domestic bovids.
Page 4, Table 1, Cattle: change myxoploydy to myxoploidy
Lines 198 and 231: explain what type I and type II loci mean
Line 287: change ZOO-FISH to Zoo-FISH
Line 514: add HSA
Author Response
Reviewer 2
The review provides a very solid overview of molecular cytogenetics of domestic bovids.
- Page 4, Table 1, Cattle: change myxoploydy to myxoploidy
We have made this change.
- Lines 198 and 231: explain what type I and type II loci mean
We have made this change in table 2.
- Line 287: change ZOO-FISH to Zoo-FISH
We have made this change throughout the article when it appears.
- Line 514: add HAS
We have made this change (the correct term is HSA).
Reviewer 3 Report
Firstly, I would like to congratulate the authors of this extensive review paper concerning the implications of molecular cytogenetics studies in farmed animals. With over 280 sources cited, the manuscript follows all developments in animal cytogenetics, with special emphasize on recent findings and technologies being currently developed and validated.
The structure of the articles looks very sound and gives valuable information in a smart and easy to read manner, and I strongly believe that the content of this 70 years review will be valuable for researchers working on animal genetics world-wide.
I have but a few suggestions/remarks:
- Please consider including a table in the manuscript in which to present the estimated incidences of genetic defects in the species that you describe. This would only highlight further the implications of your work;
- I am missing a clear sub-section at the end of the manuscript with ‘current developments and knowledge gaps’, while also highlighting future developments needed of this area;
- Subsection 3 (Lines 352-430) is extremely short, could the authors develop more the whole genome sequencing area, when it comes to animal genetics? In the last 5-6 years, a lot of publications on WGS in bovids were published, and some of these results should be better presented in the review;
- Subsection 4 (Lines 437-496), same for this section, it is to vaguely presented, especially since nowadays the PCR techniques are the norm. Please elaborate on the newly developed techniques, e.g. KASP assays;
- The number of self-citations in the text could be regarded as to high (over 70 self-citations), could you please consider removing the unnecessary/redundant self-citing sources from the manuscript?
- And finally, the conclusions, please identify 3-5 bullet-point conclusions, as of now this section is way to general and it should summarize better the implications and progresses from animal cytogenetics.
Author Response
Reviewer 3
Firstly, I would like to congratulate the authors of this extensive review paper concerning the implications of molecular cytogenetics studies in farmed animals. With over 280 sources cited, the manuscript follows all developments in animal cytogenetics, with special emphasize on recent findings and technologies being currently developed and validated.
The structure of the articles looks very sound and gives valuable information in a smart and easy to read manner, and I strongly believe that the content of this 70 years review will be valuable for researchers working on animal genetics world-wide. I have but a few suggestions/remarks:
- Please consider including a table in the manuscript in which to present the estimated incidences of genetic defects in the species that you describe. This would only highlight further the implications of your work;
Very probably the reviewer refers to the incidences of chromosome abnormalities when referring to “genetic defects”. These data have been reported in other studies (in particular in Ducos et al., 2008, Cytogenetic screening of livestock populations in Europe: an overview, Cytogenetic Genome Res 120:26–41.) where thousands on animals were studied reporting the incidence of various chromosome abnormalities in animal population (or per breeds).
In the present review we cited almost all papers which used the molecular cytogenetics (mainly the FISH-technique) to better study the various cases of bovid species found carriers of chromosome abnormalities.
- I am missing a clear sub-section at the end of the manuscript with ‘current developments and knowledge gaps’, while also highlighting future developments needed of this area;
We have added a small paragraph at the end of the paper with one more reference.
- Subsection 3 (Lines 352-430) is extremely short, could the authors develop more the whole genome sequencing area, when it comes to animal genetics? In the last 5-6 years, a lot of publications on WGS in bovids were published, and some of these results should be better presented in the review;
We have expanded this section and added 13 new references.
- Subsection 4 (Lines 437-496), same for this section, it is too vaguely presented, especially since nowadays the PCR techniques are the norm. Please elaborate on the newly developed techniques, e.g. KASP assays;
The paragraph titled "4. PCR-based methods and molecular cytogenetics" discusses the progress in detecting chromosomal abnormalities or genome stability using molecular techniques. As suggested, PCR techniques are now commonly used in farm animals, with numerous applications that enable also the identification of genetic abnormalities. However, we have chosen to focus only on the PCR techniques that can detect chromosomal abnormalities, rather than genetic abnormalities. This is because specific technologies (such as KASP or SNP) are needed to identify certain genetic abnormalities in farm animals. Including all PCR techniques in this review, which is focused on cytogenetic technique evolutions, would be non-exhaustive and not relevant to the objectives of the review. Therefore, we have chosen to limit our discussion to PCR techniques that detect chromosomal abnormalities.
- The number of self-citations in the text could be regarded as to high (over 70 self-citations), could you please consider removing the unnecessary/redundant self-citing sources from the manuscript?
Our group is the one which published the highest number of papers on FISH-mapping in bovids, especially in comparative FISH-mapping studies, thus explaining the self-citation as reported by the reviewer. It is difficult to reduce our citations when comparing them with other contributions. We prefer to keep all the citations and references we reported in our review because they are important in the present review and can be also useful for the readers.
- And finally, the conclusions, please identify 3-5 bullet-point conclusions, as of now this section is way to general and it should summarize better the implications and progresses from animal cytogenetics.
We have added what was requested.